# Metformin Use before Influenza Vaccination May Lower the Risks of Influenza and Related Complications

**DOI:** 10.3390/vaccines10101752

**Published:** 2022-10-19

**Authors:** Fu-Shun Yen, James Cheng-Chung Wei, Ying-Hsiu Shih, Chung Y. Hsu, Chih-Cheng Hsu, Chii-Min Hwu

**Affiliations:** 1Dr. Yen’s Clinic, Taoyuan 33354, Taiwan; 2Department of Allergy, Immunology & Rheumatology, Chung Shan Medical University Hospital, Taichung 40201, Taiwan; 3Institute of Medicine, Chung Shan Medical University, Taichung 40201, Taiwan; 4Graduate Institute of Integrated Medicine, China Medical University, Taichung 40402, Taiwan; 5Management Office for Health Data, China Medical University Hospital, Taichung 40201, Taiwan; 6College of Medicine, China Medical University, Taichung 40402, Taiwan; 7Graduate Institute of Biomedical Sciences, China Medical University, Taichung 40402, Taiwan; 8Institute of Population Health Sciences, National Health Research Institutes, 35 Keyan Road, Zhunan, Miaoli County 35053, Taiwan; 9Department of Health Services Administration, China Medical University, Taichung 40402, Taiwan; 10Department of Family Medicine, Min-Sheng General Hospital, Taoyuan 33024, Taiwan; 11National Center for Geriatrics and Welfare Research, National Health Research Institutes, Miaoli County 35053, Taiwan; 12Faculty of Medicine, National Yang-Ming University School of Medicine, Taipei 11121, Taiwan; 13Section of Endocrinology and Metabolism, Department of Medicine, Taipei Veterans General Hospital, Taipei 11121, Taiwan

**Keywords:** influenza, vaccination, pneumonia, mortality

## Abstract

Older adults are more likely to have influenza and respond less well to the flu vaccine. We conducted this study to investigate whether pre-influenza vaccination metformin use had an effect on influenza and relevant complications in older adults with type 2 diabetes mellitus. Propensity score matching was used to identify 28,169 pairs of metformin users and nonusers from Taiwan’s National Health Insurance Research Database from 1 January 2000 to 31 December 2018. We used Cox proportional hazards models to calculate the risks of hospitalization for influenza, pneumonia, cardiovascular disease, ventilation, and mortality between metformin users and nonusers. Compared with metformin nonusers, the aHRs (95% CI) for metformin users at risk of hospitalization for influenza, pneumonia, cardiovascular disease, invasive mechanical ventilation, death due to cardiovascular disease, and all-cause mortality were 0.60 (0.34, 1.060), 0.63 (0.53, 0.76), 0.41 (0.36, 0.47), 0.56 (0.45, 0.71), 0.49 (0.33, 0.73), and 0.44 (0.39, 0.51), respectively. Higher cumulative duration of metformin use was associated with lower risks of these outcomes than no use of metformin. This cohort study demonstrated that pre-influenza vaccination metformin use was associated with lower risks of hospitalizations for influenza, pneumonia, cardiovascular disease, mechanical ventilation, and mortality compared to metformin nonusers.

## 1. Introduction

All living things are at risk of contracting infections, and humans are no exception. Older adults have a waning immune function and multiple comorbidities. They are more prone to developing influenza, related complications, and mortality [1,2]. Population aging has increased global morbidity and mortality rates due to influenza [2]. From 1999 to 2019, the number of older adults (age ≥ 65 years) dying of influenza increased from 102,123 to 154,695 worldwide [3]. Studies have also shown that about 90% of influenza deaths occur in seniors above 60 years [4]. It is important to prevent the elderly from influenza infection, related complications, and death. Persons with type 2 diabetes mellitus (T2DM) are susceptible to influenza due to declined cellular immunity [5]. Studies have shown that T2DM patients are prone to influenza-related pneumonia, cardiovascular complications, and death [6,7].

Studies have demonstrated that the influenza vaccine significantly reduced influenza infection in healthy young adults during the pandemic of 1957 [8]. In 1960, annual influenza vaccination was recommended for people above 65 years, pregnant women, and persons with chronic debilitating diseases. Public health officials recommend annual influenza vaccination for children over 6 months, adults above 50 years, or individuals with chronic diseases (such as diabetes mellitus) [8]. Subsequent studies have shown that the influenza vaccine can significantly reduce flu infection and complications [8,9]. Observational studies also showed that influenza vaccination for patients with T2DM could reduce the risk of hospitalization and death due to influenza [10,11,12]. One systemic review has indicated that patients with T2DM may have normal humoral and cellular responses to influenza vaccination [13]. However, aging is accompanied by a decrease in the number and diversity of naïve T cells and the number of functional B cells; the elderly are often prone to chronic low-grade inflammation, leading to immune dysfunction. Therefore, the effectiveness of influenza vaccination in older adults seems to be suboptimal [9,14,15]. It is necessary to develop methods that increase the effectiveness of influenza vaccination in older adults.

Since 1995, metformin has been the drug of choice for managing diabetes mellitus. It is safe, inexpensive, and effective in glucose control [16]. Moreover, it can improve innate and adaptive immunity and mitigate infections and related complications [17]. Studies show that the mechanistic target of rapamycin (mTOR) inhibitors can improve antibody responses to flu vaccination and decrease respiratory tract infections in community-dwelling older adults [18]. Diaz et al. demonstrated that metformin could reduce B-cell intrinsic inflammation and improve the in vivo and in vitro antibody response to flu vaccines in patients with obesity and type 2 diabetes mellitus (T2DM) [19]. No study has examined whether metformin can enhance the responses of elderly patients with T2DM to influenza vaccination. Therefore, we conducted this cohort study to investigate whether metformin use before flu vaccination could reduce influenza infection, related complications, and death in patients with T2DM.

## 2. Methods

### 2.1. Data Sources

We identified participants from the full population National Health Insurance Research Database (NHIRD). In 1995, Taiwan established the National Health Insurance (NHI) program, an insurance system with the government being the single buyer. The public pays a small premium, and the government and employers pay most of the costs. By 2000, 95% of the 23 million people in the country were covered by NHI [20]. This health insurance program includes about 10,000 clinics and 450 hospitals for patient health care. Baseline information on disease diagnosis, medical procedures, and prescriptions are recorded in the NHIRD. Diagnosis is based on the International Classification of Diseases, Ninth and Tenth Revision, Clinical Modification (ICD-9/10-CM). The NHIRD is linked to the National Death Registry to provide death information. This study was performed in accordance with the Declaration of Helsinki and received approval from the Research Ethics Committee of China Medical University and Hospital [CMUH110-REC1-038(CR-1)]. The recognizable information of participants and healthcare givers was deidentified and encrypted before release to protect personal privacy. Informed consent was waived by the Research Ethics Committee.

### 2.2. Study Population

The Taiwan government provides free influenza vaccination annually for seniors ≥ 65 years from October to December. We recruited participants newly diagnosed with T2DM from 1 January 2000 to 31 December 2018, and traced them until 31 December 2019. The diagnosis of T2DM was based on ICD codes (ICD-9 code: 250, except 250.1x; ICD-10: E11. Table 1) for at least two outpatient claims or one admission record. The algorithm for using ICD codings to define T2DM was validated by a previous study in Taiwan, with an accuracy of 74.6% [21]. Participants were excluded if they were (1) below 65 or above 90 years on the date of influenza vaccination; (2) without age or sex data; (3) diagnosed with hepatic failure, type 1 diabetes mellitus, or under dialysis; (4) diagnosed with T2DM before 1 January 2000, to exclude prevalent T2DM (Figure 1).

### 2.3. Study Design and Procedures

Participants who received metformin for at least 28 days before influenza vaccination were defined as study cases and those who never received metformin during the study period served as control cases. The date of influenza vaccination (ICD-9: V04.7, V04.81; ICD-10: Z23) was defined as the index date for the study and control groups, and they were in the same calendar year. As the influenza strain can vary each year, we traced the outcomes from the date of influenza vaccination (index date) to 365 days. Some critical variables observed and compared between the study and control groups were as follows: gender, age, smoking status, overweight, alcohol-related disorders, dyslipidemia, hypertension, stroke, coronary artery disease (CAD), atrial fibrillation, heart failure, peripheral arterial disease (PAD), chronic obstructive pulmonary disease (COPD), chronic kidney disease (CKD), retinopathy, liver cirrhosis, and neoplasms diagnosed 1 year before the index date; and medications, such as sulfonylureas, thiazolidinediones, alpha-glucosidase inhibitors (AGI), dipeptidyl peptidase-4 (DPP-4) inhibitors, sodium-glucose cotransporter 2 (SGLT2) inhibitors, numbers of oral antidiabetic drugs (OAD), glucagon-like peptide-1 receptor agonists (GLP-1 RA), insulin, statins, aspirin, and nonsteroidal anti-inflammatory drugs. We also calculated the Diabetes Complication Severity Index (DCSI), Charlson Comorbidity Index (CCI) scores [22,23], the items, and the number of oral antidiabetic drugs to evaluate the severity of T2DM.

### 2.4. Outcome Measures

The main outcomes of this study were hospitalization for influenza, pneumonia, cardiovascular disease, noninvasive positive pressure ventilation (NIPPV), invasive mechanical ventilation (IMV), death due to cardiovascular disease, and all-cause mortality. We could not procure laboratory data from the NHIRD; hence, we used the ICD codings to censor these outcomes. We obtained information on death from the National Death Registry. We assessed the outcomes in the time scale of 1000 patient years and examined the cumulative incidences of the main outcomes between metformin users and nonusers during the follow-up period. To observe the outcomes of interest, we censored participants until the day of respective outcomes, death, or at the end of the follow-up time, whichever showed first.

### 2.5. Statistical Analysis

We used propensity score matching to balance the variables between metformin users and nonusers. We estimated the propensity score for every participant via non-parsimonious multivariable logistic regression. We used optimal matching by the method of greedy nearest neighbor matching and matched the comparison groups without replacement [24]. The variables of baseline characteristics, comorbidities, CCI and DCSI scores, and medications were used as independent variables; metformin use was the dependent variable. Standardized mean difference (SMD) was used to examine the dissimilarity between the matching pairs, and SMD < 0.05 was assumed as a negligible difference between the study and comparison groups.

We used the Chi-square test to compare the statistical difference of categorical variables and Student’s *t*-test to determine the statistical difference between continuous variables in the study and comparison groups. Crude and multivariable-adjusted Cox proportional hazards models were used to compare the outcomes between metformin users and nonusers. Schoenfeld’s residuals test confirmed the nonviolation of the proportional-hazards assumption. The results were shown as hazard ratios (HRs) and 95% confidence intervals (CIs). The Kaplan–Meier method and log-rank tests were used to explore the cumulative incidences of hospitalization for influenza, pneumonia, cardiovascular disease, death due to cardiovascular disease, and all-cause mortality between metformin users and nonusers during the follow-up time. We also observed the cumulative duration (28–89, 90–179, >179 days) of metformin use and risk of hospitalization for influenza, pneumonia, cardiovascular disease, death due to cardiovascular disease, and all-cause mortality compared with metformin no-use.

A two-tail *p*-value less than 0.05 was considered statistically significant. SAS (version 9.4; SAS Institute, Cary, NC, USA) was used for statistical analysis in this study.

## 3. Results

### 3.1. Participants

From 1 January 2000 to 31 December 2018, we recruited 29,255,389 patients with newly diagnosed T2DM. Of these, 917,786 received influenza vaccination, 494,071 persons received metformin, and 423,715 did not receive metformin before flu vaccination. After excluding unsuitable participants, 1: 1 propensity score matching was applied to identify 28,169 pairs of metformin users and nonusers (Figure 1). In the matched cohorts, 52.08% of participants were female; the mean (SD) age was 68.02 (3.95) years (Table 2). The follow-up time for metformin users and nonusers was 0.999 (0.041) and 0.996 (0.033) years, respectively.

### 3.2. Main Outcomes

The matched cohorts revealed that 19 (0.07%) metformin users and 31 (0.11%) metformin nonusers were hospitalized for influenza during the follow-up time (incidence rate: 0.7 vs. 1.1 per 1000 person years). The multivariable model showed that metformin users had an insignificantly lower risk of hospitalization for influenza (aHR = 0.60, 95% CI = 0.34–1.06, *p* = 0.076) than metformin nonusers (Table 2); 193 (0.69%) metformin users and 306 (1.09%) nonusers were hospitalized for pneumonia. The multivariable model showed that metformin users had significantly lower risks of hospitalization for pneumonia (aHR = 0.63, 95% CI = 0.53–0.76), cardiovascular disease (aHR = 0.41, 95% CI 0.36–0.47), invasive mechanical ventilation (aHR = 0.56, 95% CI 0.45–0.71), death due to cardiovascular disease (aHR = 0.49, 95% CI 0.33–0.73), and all-cause mortality (aHR = 0.44, 95% CI 0.39–0.51), but with an insignificant lower risk of noninvasive positive pressure ventilation (aHR = 0.75, 95% CI 0.5–1.11, *p* = 0.1543) than metformin nonusers (Table 3).

The Kaplan–Meier analysis showed that the cumulative incidence of hospitalization for influenza in metformin users was insignificantly lower than that in metformin nonusers (Log-rank test *p*-value = 0.089; Figure 2a). The cumulative incidences of hospitalization for pneumonia (Log-rank test *p*-value < 0.001), cardiovascular disease (Log-rank test *p*-value < 0.001), invasive mechanical ventilation (Log-rank test *p*-value < 0.001), death due to cardiovascular disease (Log-rank test *p*-value < 0.001), and all-cause mortality (Log-rank test *p*-value < 0.001) in metformin users were significantly lower than those in metformin nonusers (Figure 2 and Figure 3).

### 3.3. Cumulative Duration of Metformin Use

We investigated the relationship between the cumulative duration of metformin use and the risks of hospitalization for influenza, pneumonia, cardiovascular disease, invasive mechanical ventilation, death due to cardiovascular disease, and all-cause mortality (Table 4). A higher cumulative duration (28–89, ≥90 days) of metformin use was associated with a lower risk of hospitalization for influenza (aHR 2.3(1.40–4.62), 0.44(0.2–0.96)), and a higher cumulative duration (28–89, 90–179, >179 days) of metformin use was associated with further lower risks of hospitalization for pneumonia (aHR 2.29(1.83–2.87), 0.54(0.41–0.71), 0.43(0.29–0.64)), cardiovascular disease (aHR 2.3(1.97–2.7), 0.43(0.35–0.53), 0.33(0.24–0.46)), invasive mechanical ventilation (aHR 2.08(1.57–2.76), 0.45(0.31–0.65), 0.34(0.2–0.56)), death due to cardiovascular disease (aHR 1.52(0.91–2.54), 0.4(0.21–0.76), 0.4(0.19–0.83)), and all-cause mortality (aHR 1.46(1.22–1.74), 0.41(0.33–0.5), 0.38(0.3–0.47)) than no-use of metformin.

## 4. Discussion

This study demonstrated that pre-influenza vaccination metformin use was associated with lower risks of hospitalization for influenza, pneumonia, cardiovascular disease, invasive mechanical ventilation, death due to cardiovascular diseases, and all-cause mortality in older adults with type 2 diabetes mellitus. A longer cumulative duration of metformin use was associated with lower risks of these outcomes compared with no use of metformin.

Older adults and those with T2DM are more susceptible to influenza, related complications, and death due to a decline in immune function [1]. The government has recommended annual flu vaccination to attenuate the risks of influenza and related complications in older adults [8]. In influenza vaccine effectiveness studies in older adults, the placebo group received no vaccine, which could increase the susceptibility of older adults to influenza and create ethical concerns. Therefore, double-blind trials of influenza vaccines in older adults are scarce [8]. A randomized controlled trial of the influenza vaccine reported 58% vaccine efficacy against serologically confirmed influenza in healthy older adults >60 years and 57% in those >70 years. However, the number of participants in the trial was relatively small [25]. Talbot et al. investigated the effectiveness of the seasonal flu vaccine in community-dwelling older adults ≥50 years and showed that the influenza vaccine prevented 61.2% (61.9% for adults aged 50–64 years and 60.7% for adults ≥65 years) of influenza-associated hospitalizations [26]. Kwong et al. showed that the seasonal flu vaccine reduced 42% of laboratory-confirmed influenza hospitalizations in community-dwelling adults >65 years [27]. These studies revealed that the flu vaccine was effective in the elderly, but the effects might be suboptimal. Researchers attempted to change the dose and adjuvants in flu vaccines to improve their effectiveness for the elderly [28]. Wang et al. conducted a cohort study and showed that influenza vaccination was associated with a reduced risk of hospitalization for influenza in elderly diabetic patients [29]. Our study demonstrated that metformin use before flu vaccination was associated with a lower risk of hospitalization for influenza. Most patients were hospitalized for complications, and not many were tested for the influenza virus, which resulted in a low incidence rate of hospitalization for influenza, yielding statistically insignificant results in this study. More prospective studies are needed to confirm our results.

Pneumonia is one of the most common and worrisome influenza complications [30]. The influenza virus can cause viral pneumonia and contribute to other causes of pneumonia [30]. Influenza and pneumonia vaccines can reduce the risk of pneumonia. Both vaccines are recommended for older adults and those with diabetes. However, many people have not received them yet [29]. Grijalva et al. reported that flu vaccination showed lower odds of laboratory-confirmed influenza-associated pneumonia in children and adults [9]. Baxter et al. showed moderate influenza vaccine effectiveness for preventing flu-associated pneumonia and influenza hospitalizations in persons above 50 years during the flu season. They showed a 12.4% estimated vaccine effectiveness in people 50–64 years and 8.5% in people ≥ 65 years [31]. Clinical studies have shown that influenza vaccination was associated with a lower risk of hospitalization for pneumonia in persons with T2DM [10,12]. Our previous study revealed that metformin use was associated with a lower risk of hospitalization for pneumonia [32]. This study showed that metformin use in older adults before flu vaccination could reduce the risk of pneumonia and invasive mechanical ventilation. In older people with diabetes, the use of metformin before flu vaccination may reduce the risk of hospitalization for pneumonia and respiratory failure.

The potential mechanisms for metformin use before influenza vaccination and lowered risks of influenza and pneumonia are (1) Metformin can increase T cell type I interferon response and control flu viral replication via the activation of AMPK [17]. (2) Metformin can promote autophagy and phagocytosis of neutrophils to contain or kill pathogens and improve innate and adaptive host immunity [17]. (3) In vitro and in vivo studies have demonstrated that metformin can decrease intrinsic B cell inflammation, induce AMPK phosphorylation, and improve B cell response to produce antibodies [14]. (4) Metformin can prompt the formation of M2 macrophages and CD8 regulatory and memory T cells; it can lower the levels of proinflammatory cytokines and reduce the cytokine-storm-like response and exaggeration of tissue trauma [17]. Collectively, metformin may enhance host immunity to infection and lessen disease burden through several immune-promoting functions.

Cardiovascular disease is a major complication of diabetes and a serious complication of influenza [33,34]. A recent meta-analysis demonstrated that influenza vaccination could reduce 34% risk of major cardiovascular events [35]. Darvishian et al. showed that seasonal influenza vaccination could reduce the risk of laboratory-confirmed influenza in older adults with cardiovascular and respiratory diseases [33]. Modin et al. showed that recent influenza vaccination was associated with a decreased risk of cardiovascular mortality and death from acute myocardial infarction or stroke in patients with diabetes [11]. Studies have shown that metformin can reduce the risk of cardiovascular disease in patients with T2DM [36]. This study showed that metformin use before flu vaccination was associated with a lower risk of morbidity and mortality in cardiovascular disease; metformin may be an option to attenuate the risks of cardiovascular disease and mortality in older adults with T2DM receiving flu vaccines.

The vast majority of flu-related deaths occur in older adults [2,4,8]. Influenza can trigger functional decline and result in serious complications, such as pneumonia, cardiovascular diseases, and death in the elderly with chronic diseases [2,37]. One study in northern California showed that influenza vaccination decreased 4.6% of all-cause mortality in people ≥65 years [38]. Cohort studies have shown that influenza vaccination was associated with lower risks of hospitalizations for complications and mortality in adults and older people with T2DM [10,29]. This study found that metformin use was associated with a lower risk of cardiovascular and all-cause mortality in older adults who received flu vaccines. This finding may be attributable to the decreased risk of hospitalization for influenza, pneumonia, and mechanical respirator use in patients who received pre-vaccination metformin. To the best of our knowledge, our study is the first to show that pre-influenza vaccination metformin use may enhance the effectiveness of influenza vaccination and reduce the risk of hospitalization for influenza, pneumonia, cardiovascular diseases, and mortality in older people with T2DM.

This cohort study used the full population administrative dataset, which can minimize selection bias and represent the population in this country. This dataset had complete information on underlying diseases in participants and medication use. A one-year follow-up of this study may be suitable to evaluate the effectiveness of seasonal influenza vaccination and related complications. However, this research still had some limitations. First, this National Health Insurance database lacked information on immunologic tests, serology, and biochemical and hemoglobin A1C results, which hindered the precise evaluation of immune function and diabetes severity. This dataset also lacked relevant details of family history, marital status, body weight, dietary and exercise habits, educational level, smoking status, and alcohol consumption. However, we assessed obesity, alcohol use, and smoking through ICD codings. We matched the critical variables of age, gender, CCI scores, comorbidities, and prescription to achieve maximal balance for the condition between the study and comparison groups. We also matched the number and items of oral antidiabetic drugs, insulin use, and DCSI scores to balance the severity of diabetes mellitus in the study and control groups. Second, owing to no test results in this dataset, we could not use laboratory methods to define influenza, possibly leading to misclassification and underestimated outcomes. However, this limitation in the study and control groups may result in nondifferential estimation. Third, metformin use in patients with dialysis or hepatic failure showed increased susceptibility to lactic acidosis; therefore, we excluded these patients to decrease confounding bias by indications [36]. Fourth, the participants in this study were mainly Taiwanese. Different ethnic characteristics and genetics may lead to different results; therefore, the results may not be extrapolated to other races. Finally, a retrospective cohort study is prone to unknown and unmeasured confounding factors. Therefore, randomized double-blind trials are needed to verify our results.

## 5. Conclusions

Although the influenza vaccine has improved, medical authorities have emphasized that high-risk patients receive seasonal influenza vaccination. Many older adults with diabetes still die of influenza and related complications annually. Our study has demonstrated that pre-vaccination metformin use may reduce the risks of hospitalizations for influenza, pneumonia, cardiovascular disease, mechanical ventilation, and mortality. In addition to recommending regular flu and pneumonia vaccinations for older adults with diabetes, we suggest that pre-vaccination metformin use may increase the effectiveness of flu vaccines and mitigate severe complications and deaths.

## Figures and Tables

**Figure 1 vaccines-10-01752-f001:**
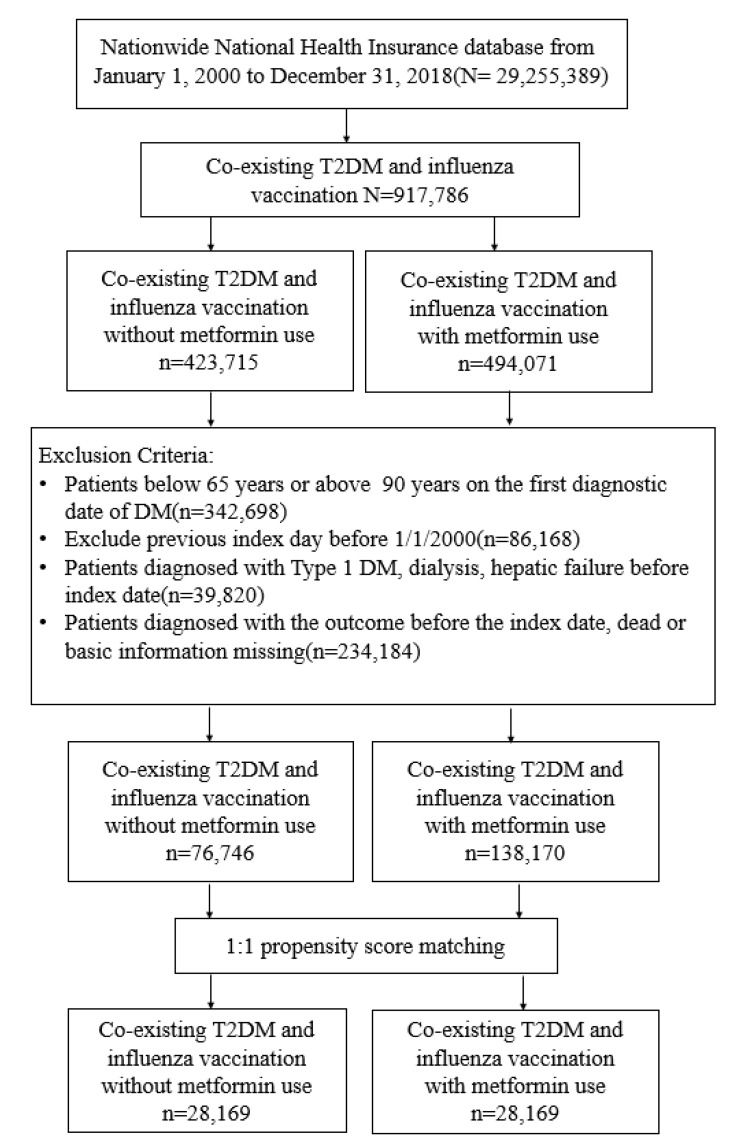
Flow chart showing the selection of study participants.

**Figure 2 vaccines-10-01752-f002:**
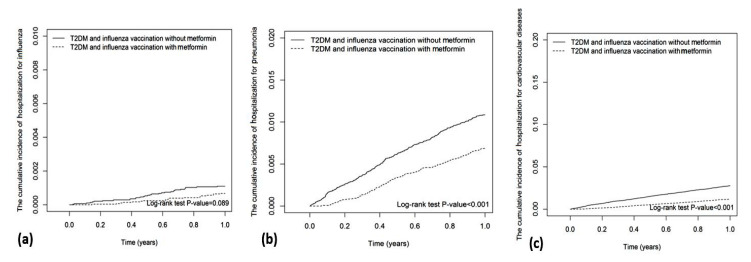
Cumulative incidences of hospitalization for (**a**) influenza, (**b**) pneumonia, and (**c**) cardiovascular disease.

**Figure 3 vaccines-10-01752-f003:**
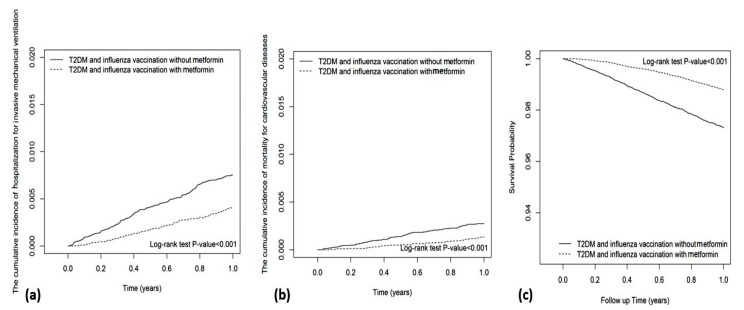
Cumulative incidences of (**a**) invasive mechanical ventilation, (**b**) death due to cardiovascular disease, (**c**) mortality.

**Table 1 vaccines-10-01752-t001:** Diseases and related ICD-9-CM, ICD-10 cm codes.

Disease	ICD-9 cm Codes	ICD-10 cm Codes
Type 2 diabetes mellitus	250.xx, except 250.1x	E11
Type 1 diabetes mellitus	250.1x	E10
Hepatic failure	570, 572.2, 572.4, 572.8	K72.00, K72.01, K72.10, K72.11, K72.90, K76.2, K72.90, K72.91, K76.7, K76.81
Dialysis	V56.0, V56.8, V45.1	Z49.31, Z49.32, Z99.2
Obesity	278.02, 783.1, V85.2, 278.00, 649.1, V77.8, V85.3, 278.01, 649.2, V45.86, V85.4	R63.5, E66.09, E66.1, E66.8, E66.9, Z13.89, E66.01, E66.2
Smoking status	305.1, 649.0, V15.82	F17.200, F17.201, F17.210, F17.220, F17.221, F17.290, F17.291, Z87.891
Alcohol-related disorders	291, 303, 305.0, 571.0-571.3, V11.3, V79.1	F10, K70.40, K70.41, K70.9
Hypertension	401–405, A26	I10, I11, I12, I13, I15, N26
Dyslipidemia	272	E71.30, E71.31, E71.32, E71.39, E75.21, E75.22, E75.23, E75.24, E75.25, E75.29, E75.3, E75.4, E75.5, E75.6, E77, E78.0, E78.1, E78.2, E78.3, E78.4, E78.5, E78.6, E78.70, E78.71, E78.72, E78.79, E78.8, E78.9
Coronary artery disease	410–414	I20, I21, I22, I24, I25.1, I25.2, I25.3, I25.4, I25.5, I25.6, I25.7, I25.81, I25.82, I25.83, I25.84, I25.89, I25.9
Chronic kidney disease	250.4x, 403.xx, 404.xx, 585.xx, 586.xx, 581.8x, 791.0x, 593.9x	E10.2, E10.65, E11.2, E11.65, E13.2, I12, I13, N03, N08, E10.21, E11.21, N05, N06, N07, N14, N15.0, N15.8, N15.9, N16, N17.1, N17.2, N18, N19
Stroke	430–438	G45.0, G45.1, G45.2, G45.3, G45.4, G45.8, G45.9, G46, I60, I61, I62, I63, I65, I66, I67.0, I67.1, I67.2, I67.3, I67.4, I67.5, I67.6, I67.7, I67.8, I67.9, I68, I69
Atrial fibrillation	427	I45.0, I45.1, I45.2, I45.3, I45.4, I45.5, I45.6
Heart failure	428	I50
Peripheral arterial disease	440.0, 440.20, 440.21, 440.22, 440.23, 440.24, 440.3, 440.4, 443.9, 443.81, 443.89	I70.2, I70.92, I75.0, I73.9
Chronic obstructive pulmonary disease	491, 492, or 496	J41, J42, J44, J43, or J44.9
Pneumonia	480–486	J12-18
Liver cirrhosis	571.5, 571.2, 571.6	K70.2, K70.30, K70.31, K74.0, K74.1, K74.2, K74.60, K74.69, K74.3, K74.4, K74.5
Cancers	140–178, 190–199, 209	C00-C63, C69-C80, C7A-C7B
Influenza	487	J09-11
Noninvasive positive pressure ventilation	93.90 and 93.91	Z99.81
Invasive mechanical ventilation	96.7	Z99.1

**Table 2 vaccines-10-01752-t002:** Matched characteristics of patients with and without metformin use before influenza vaccination.

Variables	Influenza Vaccination without Metformin		Influenza Vaccination with Metformin		SMD
	(N = 28,169)		(N = 28,169)		
	*n*	%	*n*	%	
Sex					
female	14,808	52.57	14,531	51.59	0.020
male	13,361	47.43	13,638	48.41	0.020
Age					
65–69	21,579	76.61	21,311	75.65	0.022
70–74	4480	15.90	4614	16.38	0.013
75–79	2110	7.49	2244	7.97	0.018
mean, (SD) ^†^	67.99	3.92	68.05	3.97	0.015
Obesity	696	2.47	674	2.39	0.005
Smoking status	663	2.35	686	2.44	0.005
Comorbidities					
Hypertension	21,976	78.01	21,795	77.37	0.015
Dyslipidemia	21,189	75.22	20,729	73.59	0.037
Coronary artery disease	10,163	36.08	10,067	35.74	0.007
Stroke	4984	17.69	5062	17.97	0.007
Atrial fibrillation	23	0.08	28	0.10	0.006
PAD	2486	8.83	2453	8.71	0.004
CKD	4852	17.22	4861	17.26	0.001
Retinopathy	434	1.54	478	1.70	0.012
COPD	7328	26.01	7322	25.99	0.001
Liver cirrhosis	486	1.73	519	1.84	0.009
Cancers	1771	6.29	1815	6.44	0.006
Alcohol-related disorder	1191	4.23	1249	4.43	0.010
CCI scores					
1	24,457	86.82	24,325	86.35	0.014
2–3	3150	11.18	3226	11.45	0.009
>3	562	2.00	618	2.19	0.014
DCSI scores					
0	5935	21.07	6019	21.37	0.007
1	4664	16.56	4643	16.48	0.002
≥2	17,570	62.37	17,507	62.15	0.005
Medication					
Sulfonylureas	4549	16.15	4731	16.80	0.017
Thiazolidinedione	426	1.51	500	1.78	0.021
DPP-4 inhibitor	619	2.20	710	2.52	0.021
AGI	1120	3.98	1110	3.94	0.002
Number of oral antidiabetic drugs					
1	27,337	97.05	27,143	96.36	0.039
2	817	2.90	995	3.53	0.036
≧3	15	0.05	31	0.11	0.020
Insulin	11,923	42.33	11,854	42.08	0.005
Statin	16,823	59.72	16,582	58.87	0.017
Aspirin	13,749	48.81	13,703	48.65	0.003
NSAIDs	27,772	98.59	27,677	98.25	0.027

Abbreviations: SMD, standardized mean difference; SD, standard deviation; CAD, coronary artery disease; PAD, peripheral arterial disease; CKD, chronic kidney disease; COPD, chronic obstructive pulmonary disease; CCI; Charlson Comorbidity Index; DCSI, Diabetes Complications Severity Index; DPP-4, dipeptidyl peptidase-4; AGI, alpha-glucosidase; NSAID, nonsteroidal anti-inflammatory drug. Data shown as *n* (%) or mean ± SD. ^†^: Student’s *t*-test. SMD < 0.05 indicates no significant difference between metformin users and nonusers.

**Table 3 vaccines-10-01752-t003:** Hazard ratios and 95 % confidence intervals for outcomes among the sampled patients.

Outcome	Influenza Vaccination without Metformin			Influenza Vaccination with Metformin								
	*n*	PY	IR	*n*	PY	IR	cHR	(95% CI)	*p*-Value	aHR^†^	(95% CI)	*p*-Value
Hospitalization for influenza	31	28,153	1.1	19	28,162	0.7	0.61	(0.35, 1.08)	0.0927	0.6	(0.34, 1.06)	0.0766
Hospitalization for pneumonia	306	28,002	10.9	193	28,080	6.9	0.63	(0.53, 0.75)	<0.001	0.63	(0.53, 0.76)	<0.001
Hospitalization for CVD	783	27,753	28.2	334	28,018	11.9	0.42	(0.37, 0.48)	<0.001	0.41	(0.36, 0.47)	<0.001
NIPPV	58	28,137	2.1	43	28,151	1.5	0.74	(0.5, 1.1)	0.1363	0.75	(0.5, 1.11)	0.1543
IMV	212	28,057	7.6	114	28,119	4.1	0.54	(0.43, 0.67)	<0.001	0.56	(0.45, 0.71)	<0.001
CVD mortality	78	28,129	2.8	38	28,154	1.3	0.49	(0.33, 0.72)	<0.001	0.49	(0.33, 0.73)	<0.001
All-cause mortality	758	27,796	27.3	339	28,039	12.1	0.44	(0.39, 0.5)	<0.001	0.44	(0.39, 0.51)	<0.001

Abbreviations: PY: person years; IR: incidence rate, per 1000 person years; cHR, crude hazard ratio; aHR: adjusted hazard ratio; CVD: cardiovascular diseases; NIPPV: Noninvasive positive pressure ventilation; IMV: Invasive mechanical ventilation. aHR^†^: multivariable analysis, including sex, age, obesity, smoking, comorbidities, CCI, DCSI, item, and number of oral antidiabetic drugs, insulin, statin, aspirin, and NSAIDs.

**Table 4 vaccines-10-01752-t004:** Hazard ratio for outcome for stratification by cumulative duration of metformin use.

Variables	*n*	PY	IR	cHR	(95% CI)	aHR^†^	(95% CI)
Hospitalization for influenza							
Metformin no-use	31	28,153	1.10	1.00	(Reference)	1.00	(Reference)
Cumulative duration of metformin use (days)							
28–89	11	3754	2.93	2.66	(1.34, 5.29) **	2.3	(1.14, 4.62) *
≥90	8	16,276	0.49	0.45	(0.21, 0.97) *	0.44	(0.2, 0.96) *
Hospitalization for pneumonia							
Metformin no-use	306	28,002	10.93	1.00	(Reference)	1.00	(Reference)
Cumulative duration of metformin use (days)							
28–89	106	3742	28.32	2.59	(2.08, 3.23) ***	2.29	(1.83, 2.87) ***
90–179	60	11,575	5.18	0.47	(0.36, 0.63) ***	0.54	(0.41, 0.71) ***
>179	27	4609	5.86	0.54	(0.36, 0.79) **	0.43	(0.29, 0.64) ***
Hospitalization for cardiovascular diseases							
Metformin no-use	783	27,753	28.21	1.00	(Reference)	1.00	(Reference)
Cumulative duration of metformin use (days)							
28–89	196	3139	62.44	2.21	(1.89, 2.59) ***	2.3	(1.97, 2.7) ***
90–179	101	9591	10.53	0.37	(0.3, 0.46) ***	0.43	(0.35, 0.53) ***
>179	37	3804	9.73	0.34	(0.25, 0.48) ***	0.33	(0.24, 0.46) ***
Invasive mechanical ventilation							
Metformin no-use	212	28,057	7.56	1.00	(Reference)	1.00	(Reference)
Cumulative duration of metformin use (days)							
28–89	65	3741	17.38	2.3	(1.74, 3.04) ***	2.08	(1.57, 2.76) ***
90–179	33	11,613	2.84	0.38	(0.26, 0.54) ***	0.45	(0.31, 0.65) ***
>179	16	4626	3.46	0.46	(0.28, 0.76) **	0.34	(0.2, 0.56) ***
Cardiovascular-cause mortality							
Metformin no-use	78	28,129	2.77	1.00	(Reference)	1.00	(Reference)
Cumulative duration of metformin use (days)							
28–89	19	3742	5.08	1.83	(1.11, 3.02) *	1.52	(0.91, 2.54)
90–179	11	11,635	0.95	0.34	(0.18, 0.64) ***	0.4	(0.21, 0.76) **
>179	8	4650	1.72	0.62	(0.3, 1.28)	0.4	(0.19, 0.83) *
All-cause mortality							
Metformin no-use	758	27,796	27.27	1.00	(Reference)	1.00	(Reference)
Cumulative duration of metformin use (days)							
28–89	150	3678	40.79	1.5	(1.26, 1.78) ***	1.46	(1.22, 1.74) ***
90–179	106	11,601	9.14	0.33	(0.27, 0.41) ***	0.41	(0.33, 0.5) ***
>179	83	4633	17.92	0.66	(0.52, 0.82) ***	0.38	(0.3, 0.47) ***

Abbreviations: PY: person years; IR: incidence rate, per 1000 person years; cHR, crude hazard ratio; aHR: adjusted hazard ratio. aHR^†^: multivariable analysis, including sex, age, obesity, smoking, comorbidities, CCI, DCSI, item, and number of oral antidiabetic drugs, insulin, statin, aspirin, and NSAIDs. *: *p* < 0.05, **: *p* < 0.005, ***: *p* < 0.001.

## Data Availability

Data of this study are available from the National Health Insurance Research Database (NHIRD) published by Taiwan National Health Insurance (NHI) Administration. The data utilized in this study cannot be made available in the paper, or in a public repository due to the ‘‘Personal Information Protection Act’’ executed by the Taiwan government starting from 2012. Requests for data can be sent as a formal proposal to the NHIRD Office (https://dep.mohw.gov.tw/DOS/cp-2516-3591-113.html, accessed on 15 June 2022) or by email to stsung@mohw.gov.tw.

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
