# Peer review of "Metformin Use before Influenza Vaccination May Lower the Risks of Influenza and Related Complications"

_vaccines, 2022, doi:10.3390/vaccines10101752_

Round 1

Reviewer 1 Report

The manuscript entitled “Metformin Use Before Influenza Vaccination May Lower the Risks of Influenza and Related Complications” submitted by Chih-Cheng Hsu, Chii-Min Hwu and co-workers for consideration for publication in the MDPI journal Vaccines presents the investigation of pre-influenza vaccination metformin effect on influenza and relevant complications in older adults with type diabetes mellitus. The study demonstrated that pre-influenza vaccination metformin use was associated with a lower risk of hospitalizations for influenza, pneumonia, cardiovascular disease, mechanical ventilation and mortality than metformin nonusers. The subject of the manuscript would be interesting for the possible readers of the journal Vaccines, but in my opinion, the manuscript definitely needs revision. I recommend a major revision of the manuscript before possible publication in the MDPI journal Vaccines.

1.     The components of the subject of the manuscript are widely discussed in literature, but the presented Introduction part is too short and does not present the literature's critical review. It is unacceptable and must be revised.

2.     Why are the time frameworks presented, starting from the newest to the oldest?

3.     The quality of the Figures is too low, it must be revised.

4.     Overall, there are editing mistakes, the language may be improved and polished.

5.     The scientific novelty of the presented research should be much better highlighted.

6.     Discussion should be developed.

7.     All parts from Supplementary Information should be transferred to the manuscript. 

Author Response

Thank you for reviewing our manuscript and giving us insightful recommendations. We have responsed to the comments on the attahced files.  

Reviewer 2 Report

The article submitted for review shows an interesting conclusion established by real-world data of the effect of metformin in patients who have been vaccinated against the influenza virus. This conclusion may be encouraging and opens avenues of future work for this and other types of vaccinations.

One of the limitations of the study may be that it was conducted in a single country and how the genetics of these individuals may influence the final results.

Nevertheless, it seems to me a work of high interest for the readers and that opens new lines of research. 

Author Response

Thank you for reviewing our manuscript and giving us encouragement. We have responsed to the comments on the attached file.

Round 2

Reviewer 1 Report

The revised manuscript entitled “Metformin Use Before Influenza Vaccination May Lower the Risks of Influenza and Related Complications” resubmitted by Chih-Cheng Hsu, Chii-Min Hwu and co-workers for reconsideration for publication in the MDPI journal Vaccines presents definitely higher level than the first submission. The Authors have performed all the required corrections and added additional explanations, and rewritten the manuscript accordingly. Therefore, I consider the revised manuscript suitable for publication in the MDPI journal Vaccines. My congratulations to the Authors.